# Contrastive Learning for Neural Topic Model

**Thong Nguyen**
VinAI Research
v.thongnt66@vinai.io

**Luu Anh Tuan** *
Nanyang Technological University
anhtuan.luu@ntu.edu.sg

## Abstract

Recent empirical studies show that adversarial topic models (ATM) can successfully capture semantic patterns of the document by differentiating a document with another dissimilar sample. However, utilizing that discriminative-generative architecture has two important drawbacks: (1) the architecture does not relate similar documents, which has the same document-word distribution of salient words; (2) it restricts the ability to integrate external information, such as sentiments of the document, which has been shown to benefit the training of neural topic model. To address those issues, we revisit the adversarial topic architecture in the viewpoint of mathematical analysis, propose a novel approach to re-formulate discriminative goal as an optimization problem, and design a novel sampling method which facilitates the integration of external variables. The reformulation encourages the model to incorporate the relations among similar samples and enforces the constraint on the similarity among dissimilar ones; while the sampling method, which is based on the internal input and reconstructed output, helps inform the model of salient words contributing to the main topic. Experimental results show that our framework outperforms other state-of-the-art neural topic models in three common benchmark datasets that belong to various domains, vocabulary sizes, and document lengths in terms of topic coherence.

## 1 Introduction

Topic models have been successfully applied in Natural Language Processing with various applications such as information extraction, text clustering, summarization, and sentiment analysis [1–6]. The most popular conventional topic model, Latent Dirichlet Allocation [7], learns document-topic and topic-word distribution via Gibbs sampling and mean field approximation. To apply deep neural network for topic model, Miao et al. [8] proposed to use neural variational inference as the training method while Srivastava and Sutton [9] employed the logistic normal prior distribution. However, recent studies [10, 11] showed that both Gaussian and logistic normal prior fail to capture multi-modality aspects and semantic patterns of a document, which are crucial to maintain the quality of a topic model.

To cope with this issue, Adversarial Topic Model (ATM) [10–13] was proposed with adversarial mechanisms using a combination of generator and discriminator. By seeking the equilibrium between the generator and discriminator, the generator is capable of learning meaningful semantic patterns of the document. Nonetheless, this framework has two main limitations. First, ATM relies on the key ingredient: leveraging the discrimination of the real distribution from the fake (negative) distribution to guide the training. Since the sampling of the fake distribution is not conditioned on the real distribution, it barely generates positive samples which largely preserves the semantic content of the real sample. This limits the behavior concerning the mutual information in the positive sample and the real one, which has been demonstrated as key driver to learn useful representations

---

*Corresponding author

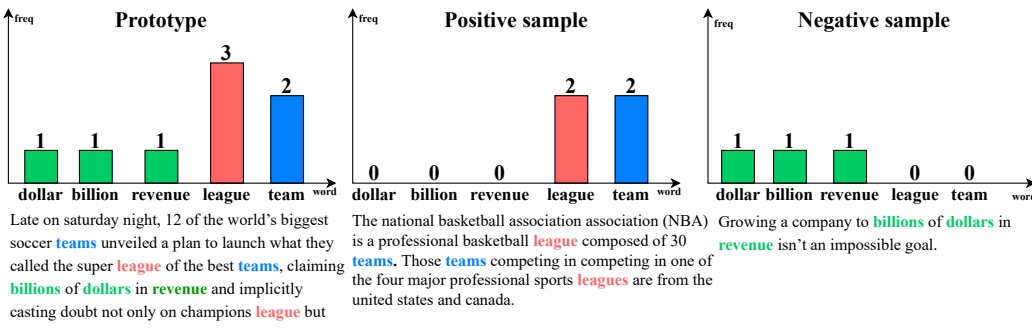

Figure 1: Illustration of a document with one positive and negative pair.

in unsupervised learning [14–18]. Second, ATM takes random samples from a prior distribution to feed to the generator. Previous work [19] has shown that incorporating additional variables, such as metadata or the sentiment, to estimate the topic distribution aids the learning of coherent topics. Relying on a pre-defined prior distribution, ATM hinders the integration of those variables.

To address the above drawbacks, in this paper we propose a novel method to model the relations among samples without relying on the generative-discriminative architecture. In particular, we formulate the objective as an optimization problem that aims to move the representation of the input (or prototype) closer to the one that shares the semantic content, i.e., positive sample. We also take into account the relation of the prototype and the negative sample by forming an auxiliary constraint to enforce the model to push the representation of the negative farther apart from the prototype. Our mathematical framework ends with a contrastive objective, which will be jointly optimized with the evidence lower bound of neural topic model.

Nonetheless, another challenge arises: how to effectively generate positive and negative samples under neural topic model setting? Recent efforts have addressed positive sampling strategies and methods to generate hard negative samples for images [20–23]. However, relevant research to adapt the techniques to neural topic model setting has been neglected in the literature. In this work, we introduce a novel sampling method that mimics the way human being seizes the similarity of a pair of documents, which is based on the following hypothesis:

**Hypothesis 1.** *The common theme of the prototype and the positive sample can be realized due to their relative frequency of salient words.*

We use the example in Fig. 1 to explain the idea of our method. Humans are able to tell the similarity of the input with positive sample due to the reason that the frequency of salient words such as *"league"* and *"teams"* is proportional to their counterpart in the positive sample. On the other hand, the separation between the input and the negative sample can be induced since those words in the input do not occur in negative sample, though they both contain words *"billions"* and *"dollars"*, which are not salient in the context of the input. Based on this intuition, our method generates the positive and negative samples for topic model by maintaining the weights of salient entries and altering those of unimportant ones in the prototype to construct the positive samples while performing the opposite procedure for the negative ones. Inherently, since our method is not depended on a fixed prior distribution to draw our samples, we are not restrained in incorporating external variables to provide additional knowledge for better learning topics.

In a nutshell, the contributions of our paper are as follows:

- We target the problem of capturing meaningful representations through modeling the relations among samples from a new mathematical perspective and propose a novel contrastive objective which is jointly optimized with evidence lower bound of neural topic model. We find that capturing the mutual information between the prototype and its positive samples provides a strong foundation for constructing coherent topics, while differentiating the prototype from the negative samples plays a less important role.

- We propose a novel sampling strategy that is motivated by human behavior when comparing different documents. By relying on the reconstructed output, we adapt the sampling to the learning process of the model, and produce the most informative samples compared with other sampling strategies.

- We conduct extensive experiments in three common topic modeling datasets and demonstrate the effectiveness of our approach by outperforming other state-of-the-art approaches in terms of topic coherence , on both global and topic-by-topic basis.

## 2   Related Work

**Neural Topic Model** (NTM) has been studied to encode a large set of documents using latent vectors. Inspired by Variational Autoencoder, NTM inherit most techniques from VAE-specific early works, such as reparameterization trick [24] and neural variational inference [25]. Subsequent works attempting to apply for topic model [9, 26, 8] focus on studying various prior distributions, e.g. Gaussian or logistic normal. Recently, researches directly target to improve topic coherence through formulating it as an optimizing objective [27], incorporating contextual language knowledge [28], or passing external information, e.g. sentiment, group of documents, as input [19]. Generating topics that are human-interpretable has become the goal of a wide variety of latest efforts.

**Adversarial Topic Model** [4] is a topic modeling approach that models the topics with GAN-based architecture. The key components in that architecture consist of a generator projecting randomly sampled document-topic distribution to gain the most realistic document-word distribution as possible and a discriminator trying to distinguish between the generated and the true sample [10, 11]. To better learn informative representations of a document, Hu et al. [12] proposed adding two cycle-consistent constraints to encourage the coordination between the encoder and generator.

**Contrastive Framework and Sampling Techniques** There are various efforts studying contrastive method to learn meaningful representations. For visual information, contrastive framework is applied for tasks such as image classification [29, 30], object detection [31–33], image segmentaion [34–36], etc. Other applications different from image include adversarial training [37–39], graph [40–43], and sequence modeling [44–46]. Specific positive sampling strategies have been proposed to improve the performance of contrastive learning, e.g. applying view-based transformations that preserve semantic content in the image [22, 17, 18]. On the other hand, there is a recent surge of interest in studying negative sampling methods. Chuang et al. [20] propose a debiasing method which is to correct the fact in false negative samples. For object detection, Jin et al. [47] employ temporal structure of video to generate negative examples. Although widely studied, little effort has been made to adapt contrastive techniques to neural topic model.

In this paper, we re-formulate our goal of learning document representations in neural topic model as a contrastive objective. The form of our objective is mostly related to Robinson et al. [21]. However, there are two key differences: (1) As they use the weighting factor associated with the impact of negative sample as a tool to search for the distribution of hard negative samples, we consider it as an adaptive parameter to control the impact of the positive and negative sample on the learning. (2) We regard the effect of positive sample as the main driver to achieve meaningful representations, while they exploit the impact of negative ones. Our approach is more applicable to topic modeling, as proven in the investigation into human behavior of distinguishing among documents.

## 3   Methodology

### 3.1   Notations and Problem Setting

In this paper, we focus on improving the performance of neural topic model (NTM), measured via topic coherence. NTM inherits the architecture of Variational Autoencoder, where the latent vector is taken as topic distribution. Suppose the vocabulary has $V$ unique words, each document is represented as a word count vector $\mathbf{x} \in \mathbb{R}^V$ and a latent distribution over $T$ topics: $\mathbf{z} \in \mathbb{R}^T$. NTM assumes that $z$ is generated from a prior distribution $p(\mathbf{z})$ and x is generated from the conditional distribution over the topic $p_\phi(\mathbf{x}|\mathbf{z})$ by a decoder $\phi$. The aim of model is to infer the document-topic distribution given the word count. In other words, it must estimate the posterior distribution $p(\mathbf{z}|\mathbf{x})$, which is approximated by the variational distribution $q_\theta(\mathbf{z}|\mathbf{x})$ modelled by an encoder $\theta$. NTM is trained by minimizing the following objective

$$\mathcal{L}_{\text{VAE}}(\mathbf{x}) = -\mathbb{E}_{q_\theta(\mathbf{z}|\mathbf{x})}[\log p_\phi(\mathbf{x}|\mathbf{z})] + \mathbb{KL}[q_\theta(\mathbf{z}|\mathbf{x})||p(\mathbf{z})] \tag{1}$$

---

**Algorithm 1** Approximate $\beta$

---

**Input:** Dataset $\mathcal{D} = \{\mathbf{x}_i\}_{i=1}^N$, model parameter $\theta$, model $f$, total training steps $T$
 1: Randomly pick a batch of $L$ samples from the training set
 2: **for** each sample $\mathbf{x}_l$ in the chosen batch **do**
 3:     Draw the negative sample $\mathbf{x}_l^-$ and a positive sample $\mathbf{x}_l^+$
 4:     Obtain the latent distribution associated with the drawn samples: $\mathbf{z}_l^- = f(\mathbf{x}_l^-)$, $f(\mathbf{x}_l^+) = \mathbf{z}_l^+$
 5:     Obtain the candidate $\beta$ value with $\gamma_l = (\mathbf{z} \cdot \mathbf{z}^+)/(\mathbf{z} \cdot \mathbf{z}^-)$.
 6: **end for**
 7: Initialize $\beta$ as the mean of the candidate list $\beta_0 = \frac{1}{L} \cdot \sum_{l=1}^L \gamma_l$
 8: **for** $t = 1$ to $T$ **do**
 9:     Train the model with $\beta_t = \frac{1}{2} - \frac{1}{T} \left| \frac{T}{2} - t \right| + \beta_0$
10: **end for**

---

## 3.2  Contrastive objective derivation

Let $\mathcal{X} = \{\mathbf{x}\}$ denote the set of document bag-of-words. Each vector $\mathbf{x}$ is associated with a negative sample $\mathbf{x}^-$ and a positive sample $\mathbf{x}^+$. We assume a discrete set of latent classes $\mathcal{C}$, so that $(\mathbf{x}, \mathbf{x}^+)$ have the same latent class while $(\mathbf{x}, \mathbf{x}^-)$ does not. In this work, we choose to use the semantic dot product to measure the similarity between prototype $\mathbf{x}$ and the drawn samples.

Our goal is to learn a mapping function $f_\theta : \mathbb{R}^V \to \mathbb{R}^T$ of the encoder $\theta$ which transforms $\mathbf{x}$ to the latent distribution $\mathbf{z}$ ($\mathbf{x}^-$ and $\mathbf{x}^+$ are transformed to $\mathbf{z}^-$ and $\mathbf{z}^+$, respectively). A reasonable mapping function must fulfill two qualities: (1) $\mathbf{x}$ and $\mathbf{x}^+$ are mapped onto nearby positions; (2) $\mathbf{x}$ and $\mathbf{x}^-$ are projected distantly. Regarding goal (1) as the main objective and goal (2) as the constraint enforcing the model to learn the relations among dissimilar samples, we specify the constrained optimization problem, in which $\epsilon$ denotes the strength of the constraint

$$\max_\theta \mathbb{E}_{\mathbf{x} \sim \mathcal{X}}(\mathbf{z} \cdot \mathbf{z}^+) \quad \text{subject to } \mathbb{E}_{\mathbf{x} \sim \mathcal{X}}(\mathbf{z} \cdot \mathbf{z}^-) < \epsilon \tag{2}$$

Rewriting Eq. 2 as a Lagragian under KKT conditions [48, 49], we attain:

$$\mathcal{F}(\theta, \mathbf{x}, \mathbf{x}^+, \mathbf{x}^-) = \mathbb{E}_{\mathbf{x} \sim \mathcal{X}}(\mathbf{z} \cdot \mathbf{z}^+) - \alpha \cdot [\mathbb{E}_{\mathbf{x} \sim \mathcal{X}}(\mathbf{z} \cdot \mathbf{z}^-) - \epsilon] \tag{3}$$

where the positive KKT multiplier $\alpha$ is the regularisation coefficient that controls the effect of the negative sample on training. Eq. 3 can be derived to arrive at the weighted-contrastive loss.

$$\mathcal{F}(\theta, \mathbf{x}, \mathbf{x}^+, \mathbf{x}^-) \geq \mathcal{L}_{\text{cont}}(\theta, \mathbf{x}, \mathbf{x}^+, \mathbf{x}^-) = \mathbb{E}_{\mathbf{x} \sim \mathcal{X}} \left[ \log \frac{\exp(\mathbf{z} \cdot \mathbf{z}^+)}{\exp(\mathbf{z} \cdot \mathbf{z}^+) + \beta \cdot \exp(\mathbf{z} \cdot \mathbf{z}^-)} \right] \tag{4}$$

where $\alpha = \exp(\beta)$. The full proof of (4) can be found in the Appendix. Previous works [39, 35, 40, 29, 20, 50] consider the positive and negative sample equally likely as setting $\beta = 1$. In this paper, we leverage different values of $\beta$ to guide the model concentration on the sample which is distinct from the input. In consequence, a reasonable value of $\beta$ will provide a clear separation among topics in the dataset. We demonstrate our procedure to estimate $\beta$ in the following section.

## 3.3  Controlling the effect of negative sample

When choosing value of $\beta$, we need to answer the following questions: (1) What impact does $\beta$ have on the process of training? and (2) Is it possible to design a procedure which is data-oriented to approximate $\beta$?

**Understanding the impact of $\beta$**   To exemplify point (1), we study the impact of $\beta$ on the contrastive loss presented in Section 3.2. The gradient of the contrastive loss (4) with respect to the latent distribution $\mathbf{z}$ would be:

$$\frac{\delta \mathcal{L}_{\text{cont}}}{\delta \mathbf{z}} = \mathbb{E}_{\mathbf{x} \sim \mathcal{X}} \left[ \frac{(\mathbf{z}^+ - \mathbf{z}^-) \cdot \exp(\mathbf{z} \cdot \mathbf{z}^-)}{\exp(\mathbf{z} \cdot \mathbf{z}^+)/\beta + \exp(\mathbf{z} \cdot \mathbf{z}^-)} \right] \tag{5}$$

This derivation confirms the proportionality of the gradient norm with respect to $\beta$. As the training progresses, the update step must be carefully controlled to avoid bouncing around the minimum or getting stuck in local optima.

**Adaptive scheduling** We leverage the adaptive approach to construct a data-oriented procedure to estimate $\beta$. Initially, the neural topic model will consider the representation of each document equally likely. The relation of the similarity of the positive and the prototype to the one of the negative and the prototype can provide us with a starting viewpoint of the model. Concretely, we store that information in the initialized value of $\beta$ which is estimated with the formula $\beta_0 = \mathbb{E}_{\mathbf{x} \sim \mathcal{X}} \left[ (\mathbf{z} \cdot \mathbf{z}^+)/(\mathbf{z} \cdot \mathbf{z}^-) \right]$.

After intialisation, to accommodate to the model learning, we continue to adopt an adaptive strategy which keeps updating value of $\beta$ according to the triangle scheduling procedure: $\beta_t = \frac{1}{2} - \frac{1}{T} \left| \frac{T}{2} - t \right| + \beta_0$. We summarize the detail of choosing $\beta$ in Algo. 1.

### 3.4 Word-based Sampling Strategy

Here we provide a technical motivation and details of our sampling method. To choose a sample which has the same underlying topic with the input, it is reasonable to filter out $M$ topics which hold large values in the document-topic distribution, as they are considered to be important by the neural topic model. Subsequently, the procedure will draw salient words in each of the topic that will contribute the weights to the drawn samples. We call this strategy as the topic-based sampling strategy.

However, as shown in [8], the process of topic choosing is sensitive to the training performance and it is challenging to determine the optimal topic number represented for every single input. Miao et al [8] implemented a stick breaking procedure to specifically predict number of topics for each document. Their strategy demands approximating the likelihood increase for each decision of breaking the stick, in other word adding the number of topic that the document denotes. Since their process takes up a considerable amount of computation, we propose a simpler approach which is word-based to draw both positive and negative samples.

For each document with its associated word count vector $\mathbf{x} \in \mathcal{X}$, we form the *tf-idf* representation $\mathbf{x}^{\text{tfidf}}$. Then, we feed x to the neural topic model to obtain the latent vector $\mathbf{z}$ and the reconstructed document $\mathbf{x}^{\text{recon}}$. Our word-based sampling strategy is illustrated in Fig. 2.

**Negative sampling** We select $k$ tokens $N = \{n_1, n_2, \ldots, n_k\}$ that have the highest *tf-idf* scores. We hypothesize that these words mainly contribute to the topic of the document. By substituting weights of chosen tokens in the original input $\mathbf{x}$ with the weights of the reconstructed representation $\mathbf{x}^{\text{recon}}$: $\mathbf{x}_{n_j}^- = \mathbf{x}_{n_j}^{\text{recon}}, j \in \{1,..,k\}$, we enforce the negative samples $\mathbf{x}^-$ to have the main content deviated from the original input $\mathbf{x}$.

Note that since the model improves its reconstruction ability as training progresses, the weights of salient words from the reconstructed output approach those from the original input (but not equal). The model should take a more careful learning step to adapt to this situation. As the negative sample controlling factor $\beta$ decays its value when converging to the final training step, due to our adaptive scheduling approach aforementioned in section 3.3, it is able to adapt to this phenomenon.

**Positive sampling** Contrary to the negative case, we select $k$ tokens possessing the lowest *tf-idf* scores $P = \{p_1, p_2, \ldots, p_k\}$. We obtain the positive sample which bears a resembling theme to the original input by assigning weights of the chosen tokens in $\mathbf{x}^{\text{recon}}$ to their counterpart in $\mathbf{x}^+$ through $\mathbf{x}_{p_j}^+ = \mathbf{x}_{p_j}^{\text{recon}}, j \in \{1,..,k\}$. This forms a valid positive sampling procedure since modifying weights of insignificant tokens retains the salient topics in the source document.

### 3.5 Training objective

**Joint objective** We jointly combine the goal of reconstructing the original input, matching the approximate with the true posterior distribution, with the contrastive objective specified in section 3.2.

$$
\begin{aligned}
\mathcal{L}(\mathbf{x}, \theta, \phi) = &-\mathbb{E}_{\mathbf{z} \sim q(\mathbf{z}|\mathbf{x})} \left[ \log(p_\theta(\mathbf{x}|\mathbf{z})) + \mathbb{KL}(q_\theta(\mathbf{z}|\mathbf{x})||p(\mathbf{z})) \right] \\
&- \mathbb{E}_{\mathbf{z} \sim q(\mathbf{z}|\mathbf{x})} \left[ \log \frac{\exp(\mathbf{z} \cdot \mathbf{z}^+)}{\exp(\mathbf{z} \cdot \mathbf{z}^+) + \beta \cdot \exp(\mathbf{z} \cdot \mathbf{z}^-)} \right]
\end{aligned}
\tag{6}
$$

We summarize our learning procedure in Algorithm 2.

Figure 2: Our sampling strategy.

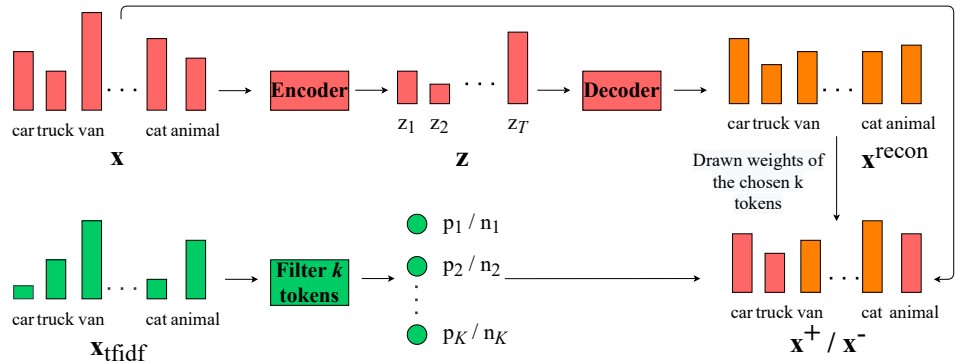

---

**Algorithm 2** Contrastive Neural Topic Model

---

**Input:** Dataset $\mathcal{D} = \{\mathbf{x}^i_{\text{tfidf}}, \mathbf{x}^i_{\text{BOW}}\}^N_{i=1}$, model parameter $\theta$, model $f$, push-pull balancing factor $\alpha$, contrastive controlling weight $\gamma$

1: **repeat**
2:     **for** $i = 1$ to $N$ **do**
3:         Compute $\mathbf{z}^i$, $\mathbf{x}^i_{\text{recon}}$ from $\mathbf{x}^i_{\text{BOW}}$;
4:         Obtain top-k indices of words with smallest *tf-idf* weights $K_{\text{pos}} = \{p_1, p_2, ..., p_k\}$;
5:         Sample $\mathbf{x}^i_{\text{pos}}$ from $K^i_{\text{pos}}$ and $\mathbf{x}^i_{\text{recon}}$;
6:         Obtain top-k indices of words with largest *tf-idf* weights $K_{\text{neg}} = \{n_1, n_2, ..., n_k\}$;
7:         Sample $\mathbf{x}^i_{\text{neg}}$ from $K^i_{\text{neg}}$ and $\mathbf{x}^i_{\text{recon}}$;
8:     **end for**
9:     Compute the loss function $\mathcal{L}$ defined in Eq. 6;
10:    Update $\theta$ by gradients to minimize the loss;
11: **until** the training converges

---

# 4 Experimental Setting

In this section, we provide the experimental setups of our conducted experiments to evaluate the performance of our proposed method. We provide the statistics summary of the datasets in Appendix.

## 4.1 Datasets

We conduct our experiments on three readily available datasets that belong to various domains, vocabulary sizes, and document lengths:

- **20Newsgroups (20NG)** dataset [51] consists of about 18000 documents, each document is a newsgroup post and associated with a newsgroup label (for example, `talk.politics.misc`). Following Huynh et al. [52], we preprocess the dataset to remove stopwords, words possessing length equal to 1, and get rid of words whose frequency is less than 100. We conduct the dataset split with 48%, 12%, 40% for training, validation, and testing, respectively.

- **Wikitext-103 (Wiki)** [53] is a version of WikiText dataset, which includes about 28500 articles from Good and Featured section on Wikipedia. We follow the preprocess, keep the top 20000 words as in [53], and use the train/dev/test split of 70%, 15%, and 15%.

- **IMDb movie reviews (IMDb)** [54] has 50000 movie reviews for analytics. Each review in the corpus is connected with a sentiment label, which we use as the external variable for our topic model. Respectively, we apply the dataset split of 50%, 25%, 25% for training, validation, and testing.

For evaluation measure, we use the Normalized Mutual Pointwise Information (NPMI) since this strongly correlates with human judgement and is popularly applied to verify the topic quality [28]. For text classification, we use the F1-score as the evaluation metric.

Table 1: Results measured in NPMI of neural topic models

| | **20NG** | | **IMDb** | | **Wiki** | |
|---|---|---|---|---|---|---|
| | $T = 50$ | $T = 200$ | $T = 50$ | $T = 200$ | $T = 50$ | $T = 200$ |
| **NTM** [27] | $0.283 \pm 0.004$ | $0.277 \pm 0.003$ | $0.170 \pm 0.008$ | $0.169 \pm 0.003$ | $0.250 \pm 0.010$ | $0.291 \pm 0.009$ |
| **W-LDA** [13] | $0.279 \pm 0.003$ | $0.188 \pm 0.001$ | $0.136 \pm 0.007$ | $0.095 \pm 0.003$ | $0.451 \pm 0.012$ | $0.308 \pm 0.007$ |
| **BATM** [11] | $0.314 \pm 0.003$ | $0.245 \pm 0.001$ | $0.065 \pm 0.008$ | $0.090 \pm 0.004$ | $0.336 \pm 0.010$ | $0.319 \pm 0.005$ |
| **SCHOLAR** [19] | $0.319 \pm 0.007$ | $0.263 \pm 0.002$ | $0.168 \pm 0.002$ | $0.140 \pm 0.001$ | $0.429 \pm 0.011$ | $0.446 \pm 0.009$ |
| **SCHOLAR + BAT** [28] | $0.324 \pm 0.006$ | $0.272 \pm 0.002$ | $0.182 \pm 0.002$ | $0.175 \pm 0.003$ | $0.446 \pm 0.010$ | $0.455 \pm 0.007$ |
| **Our model -** $k = 1$ | $0.327 \pm 0.006$ | $0.274 \pm 0.003$ | $0.191 \pm 0.007$ | $0.185 \pm 0.003$ | $0.455 \pm 0.012$ | $0.450 \pm 0.008$ |
| **Our model -** $k = 5$ | $0.328 \pm 0.004$ | $0.277 \pm 0.003$ | $0.195 \pm 0.008$ | $0.187 \pm 0.001$ | $0.465 \pm 0.012$ | $0.456 \pm 0.004$ |
| **Our model -** $k = 15$ | $\mathbf{0.334 \pm 0.004}$ | $\mathbf{0.280 \pm 0.003}$ | $\mathbf{0.197 \pm 0.006}$ | $\mathbf{0.188 \pm 0.002}$ | $\mathbf{0.497 \pm 0.009}$ | $\mathbf{0.478 \pm 0.006}$ |

## 4.2 Baselines

We compare our method with the following state-of-the-art neural topic models of diverse styles:

- **NTM** [27] a Gaussian-based neural topic model proposed by (Miao et al., 2017) inheriting the VAE architecture and utilizing neural variational inference for training.

- **SCHOLAR** [19] a VAE-based neural topic model learning with logistic normal prior and is provided with a method to incorporate external variables.

- **SCHOLAR + BAT** [28] a version of SCHOLAR model trained using knowledge distillation where BERT model as a teacher provides contextual knowledge for its student, the neural topic model.

- **W-LDA** [13] a topic model which takes form of a Wasserstein auto-encoder with Dirichlet prior approximated by minimizing Maximum Mean Discrepancy.

- **BATM** [11] a neural topic model whose architecture is inspired by Generative Adversarial Network. We use the version trained with bidirectional adversarial training method and the architecture consisting of 3 components: encoder, generator, and discriminator.

## 5 Results

### 5.1 Topic coherence

**Overall basis** We evaluate our methods both at $K = 50$ and $K = 200$. For each topic, we follow previous works [28, 10, 19] to pick the top 10 words, measure its NPMI measure and calculate in the average value. As shown in Tab. 1, our method achieves the best topic coherence on three benchmark datasets. We surpass the baseline SCHOLAR [19], its version trained with distilled knowledge SCHOLAR + BAT [28], and other state-of-the-art neural topic models in both cases of $K = 50$ and $K = 200$. We also establish the robustness of our improvement by conducting experiments on 5 runs with different random seeds and recording the mean and standard deviation. This confirms that the contrastive framework promotes the overall quality of generated topics.

Figure 3: (left) Jensen-Shannon for aligned topic pairs of SCHOLAR and our model. (right) The number of aligned topic pairs which our model improves upon SCHOLAR model

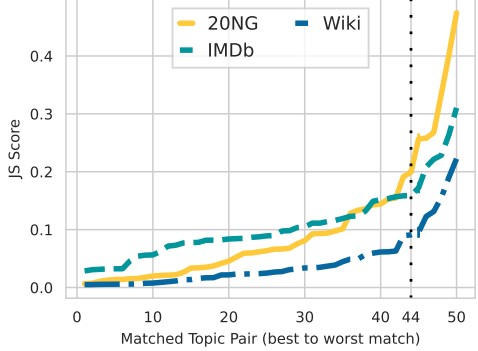
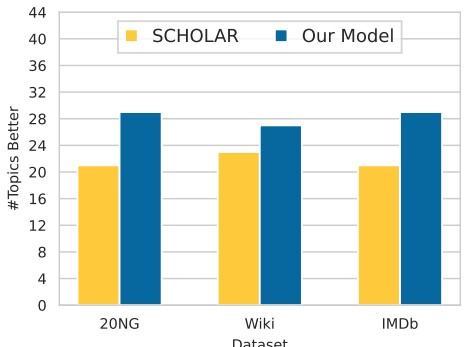

Table 3: Ablation studies

| | 20NG | | IMDb | | Wiki | |
|---|---|---|---|---|---|---|
| | $T = 50$ | $T = 200$ | $T = 50$ | $T = 200$ | $T = 50$ | $T = 200$ |
| **Our method** | **0.334 ± 0.004** | **0.280 ± 0.003** | **0.197 ± 0.006** | **0.190 ± 0.002** | **0.497 ± 0.009** | **0.478 ± 0.006** |
| - w/o positive sampling | 0.320 ± 0.004 | 0.272 ± 0.002 | 0.187 ± 0.006 | 0.182 ± 0.007 | 0.452 ± 0.012 | 0.448 ± 0.009 |
| - w/o negative sampling | 0.331 ± 0.002 | 0.277 ± 0.002 | 0.195 ± 0.008 | 0.188 ± 0.003 | 0.474 ± 0.010 | 0.468 ± 0.007 |

**Topic-by-topic basis** To further evaluate the performance of our method, we proceed to individually compare each of our topics with the aligned topic produced by the baseline neural topic model. Following Hoyle et al. [28], we use a variant of competitive linking to greedily approximate the optimal weight of the bipartite graph matching. Particularly, a bipartite graph is constructed by linking the topics of our model and the baseline one. The weight of each link is represented as the Jensen-Shannon (JS) divergence [55, 56] between two topics. We iteratively choose the pair according to its lowest JS score, dispense those two topics from the topic list, and repeat until the JS score surpasses a certain threshold. Fig. 3 (left) shows the aligned scores for three benchmark corpora. Using visual inspection, we decide to choose the most aligned 44 topic pairs to conduct the comparison. As shown in Fig. 3 (right), our model has more topics with higher NPMI score than the baseline model. This means that our model not only generates better topics on average but also on the topic-by-topic basis.

## 5.2 Text classification

In order to compare the extrinsic predictive performance, we use document classification as the downstream task. We collect the latent vectors inferred by neural topic models in $K = 50$ and train a Random Forest with the number of decision trees as 10 and the maximum depth as 8 to predict the class of each document. We pick *IMDb* and *20NG* for our experiment. Our method surpasses other neural topic models on the downstream text classification with significant gaps, as shown in Tab. 2.

Table 2: Text classification employing the latent distribution predicted by neural topic models.

| Model | 20NG | IMDb |
|---|---|---|
| **BATM** [11] | 30.8 | 66.0 |
| **SCHOLAR** [19] | 52.9 | 83.4 |
| **SCHOLAR + BAT** [28] | 32.2 | 73.1 |
| **Our model** | **54.4** | **84.2** |

## 5.3 Ablation Study

To verify the efficiency mimicking the human behavior in learning topic by grasping the commonalities, we train our methods under the besting setting with ($k = 15$, with word-based sampling), but with two different objectives: (1) Without positive sampling: model captures semantic pattern by only distinguishing the input and the negative sample; (2) Without negative sampling: model learns the semantic pattern by solely minimizing the similarity the input with the positive sample. Tab. 3 demonstrates losing one of the two views in contrastive framework degrades the quality of the topics. We include the optimizing objective for the two approaches in the Appendix. Remarkably, it is interesting that removing the negative objective influences less than for the positive one. This reconfirms the soundness of our approach to focus on the effect of positive sample, which takes inspiration from human perspective.

# 6 Analysis

## 6.1 Effect of adaptive controlling parameter

We then show the relation between $\beta$, which controls the impact of our constraint, and the topic coherence measure in Fig. 4. As shown in the figure, adaptive weight exhibits consistent superiority over manually tuned constant parameter. We elaborate our high performance on the triangle scheduling that brings the self-adjustment in different training stages.

## 6.2 Random Sampling Strategy

In this section, we demonstrate the effectiveness of our random sampling strategy. We compare our performance with two other methods: (1) 0-sampling: we replace the weights of $k$ chosen tokens in

Table 4: Results of different sampling method

| | 20NG | | IMDb | | Wiki | |
|---|---|---|---|---|---|---|
| | $T = 50$ | $T = 200$ | $T = 50$ | $T = 200$ | $T = 50$ | $T = 200$ |
| **0-sampling** | $0.269 \pm 0.003$ | $0.231 \pm 0.001$ | $0.171 \pm 0.005$ | $0.172 \pm 0.002$ | $0.448 \pm 0.008$ | $0.429 \pm 0.007$ |
| **Random sampling** | $0.321 \pm 0.005$ | $0.273 \pm 0.001$ | $0.183 \pm 0.002$ | $0.177 \pm 0.001$ | $0.460 \pm 0.012$ | $0.462 \pm 0.003$ |
| **Topic-based sampling -** $T = 1$ | $0.313 \pm 0.004$ | $0.270 \pm 0.005$ | $0.189 \pm 0.002$ | $0.172 \pm 0.002$ | $0.467 \pm 0.012$ | $0.464 \pm 0.002$ |
| **Topic-based sampling -** $T = 3$ | $0.322 \pm 0.005$ | $0.268 \pm 0.002$ | $0.181 \pm 0.006$ | $0.170 \pm 0.007$ | $0.450 \pm 0.013$ | $0.461 \pm 0.008$ |
| **Topic-based sampling -** $T = 5$ | $0.319 \pm 0.001$ | $0.273 \pm 0.002$ | $0.176 \pm 0.007$ | $0.170 \pm 0.003$ | $0.472 \pm 0.007$ | $0.444 \pm 0.006$ |
| **Our method** | $\mathbf{0.334 \pm 0.004}$ | $\mathbf{0.280 \pm 0.003}$ | $\mathbf{0.197 \pm 0.006}$ | $\mathbf{0.188 \pm 0.002}$ | $\mathbf{0.497 \pm 0.009}$ | $\mathbf{0.478 \pm 0.006}$ |

Figure 4: The influence of adaptive controlling parameter $\beta$ on topic coherence measure

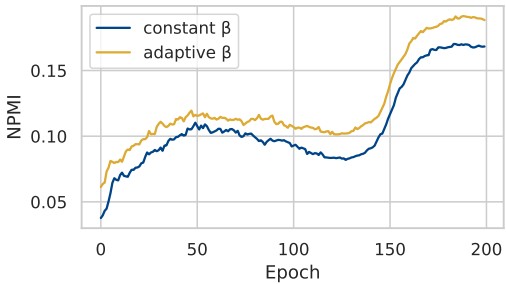

the BoW with 0; (2): we create the negative samples by drawing other documents from the dataset, then extracting the topic vector of each document; we do not perform positive sampling in this variant. (3) Topic-based sampling: the sampling strategy we discussed in section 3.4, we experiment with varying choices of $T$. As shown in Tab. 4, our sampling method consistently outperforms other strategies by a large margin. This confirms our hypothesis that topic-based sampling is vulnerable to drawing insufficient or redundant topics and might harm the performance.

In addition, to further evalute the statistical significance of our outperforming over traditional random sampling method, we conduct significance testing and report p-value in Tab. 5. As it can be seen, all of the p-values are smaller than 0.05, which proves the statistical significance in the improvement of our method against traditional contrastive learning.

### 6.3 Importance Measure

Our word-based sampling strategy employs *tf-idf* measure to determine important and unimportant words that have values to be superseded to form positive and negative samples.

To have a fair judgement, we also conduct experiments with two other complex sampling methods using Principal Component Analysis (PCA) or Singular Value Decomposition (SVD). Specifically, we decompose the reconstructed and original input vectors into singular values and then replace the largest/smallest singular values of the input with the largest/smallest ones of the reconstructed to obtain negative/positive samples, respectively. For SVD, we choose $k = 15$ largest/smallest values for substitution whereas for PCA, we decompose the input vector onto 50-d space in order to make it similar to the latent space of neural topic model (number of topics $T = 50$) and proceed to substitute $k = 15$ largest/smallest values as in SVD. We conducted our experiments on 3 datasets IMDb, 20NG, and Wiki with $T = 50$, and reported the results (NPMI) in Tab. 6.

As it can be obviously seen, despite its simplicity, *tf-idf*-based sampling method outperforms other complicated sampling methods in our tasks.

### 6.4 Case Studies

We randomly extract sample topic in each of three datasets to study the quality of the generated topics and show the result in Tab. 7. Generally, the topic words generated by our model tends to concentrate on the main topic of the document. For example, in *20NG* dataset, it can be seen that our words tend to concentrate on the topic related to cryptography (*encryption*, *crypto*, etc.) and computer hardware (*chip*, *wiretap*, *clipper*, etc.), rather than political words, e.g. *bush* and *clinton* generated

Table 5: Significance Testing results, reporting p-value

| Number of Topics | 20NG | IMDb | Wiki |
|:---:|:---:|:---:|:---:|
| $T = 50$ | 0.0140 | 0.0291 | 0.0344 |
| $T = 200$ | 0.0494 | 0.0012 | 0.0156 |

Table 6: Results when employing various importance measures

| Metrics | IMDb | 20NG | Wiki |
|:---:|:---:|:---:|:---:|
| PCA | $0.184 \pm 0.004$ | $0.325 \pm 0.003$ | $0.481 \pm 0.005$ |
| SVD | $0.181 \pm 0.004$ | $0.313 \pm 0.003$ | $0.476 \pm 0.014$ |
| *tf* | $0.196 \pm 0.003$ | $0.332 \pm 0.006$ | $0.495 \pm 0.008$ |
| *idf* | $0.193 \pm 0.001$ | $0.334 \pm 0.004$ | $0.490 \pm 0.009$ |
| *tf-idf* | $\mathbf{0.197 \pm 0.006}$ | $\mathbf{0.334 \pm 0.004}$ | $\mathbf{0.497 \pm 0.009}$ |

Table 7: Some example topics on three datasets *20NG*, *Wiki*, and *IMDb*

| Dataset | Method | NPMI | Topic |
|:---:|:---:|:---:|:---:|
| 20NG | **SCHOLAR** | 0.259 | max bush clinton crypto pgp clipper nsa announcement air escrow |
| | **Our model** | 0.543 | crypto clipper encryption nsa escrow wiretap chip proposal warrant secure |
| Wiki | **SCHOLAR** | 0.196 | airlines boeing vehicle manufactured flight skiing airline ski engine alpine |
| | **Our model** | 0.564 | skiing ski alpine athletes para paralympic nordic olympic paralympics ipc |
| IMDb | **SCHOLAR** | 0.145 | hong chinese kong imagery japanese rape lynch torture violence disturbing |
| | **Our model** | 0.216 | hong chinese kong japan fairy japanese sword martial fantasy magical |

by SCHOLAR model. Our generated topics in Wiki is more focused on *skiing*, while SCHOLAR's topic comprises of traffic terms such as *vehicle*, *boeing*, and *engine*. Similarly, the topic words in *IMDb* generated by our model mainly reflects the theme of Fantasy movie in *japan*, *chinese*, and *hong kong*, while not including off-topic words such as *torture* and *disturbing* which were generated by SCHOLAR model.

## 7 Conclusion

In this paper, we propose a novel method to help neural topic model learn more meaningful representations. Approaching the problem with a mathematical perspective, we enforce our model to consider both effects of positive and negative pairs. To better capture semantic patterns, we introduce a novel sampling strategy which takes inspiration from human behavior in differentiating documents. Experimental results on three common benchmark datasets show that our method outperforms other state-of-the-art neural topic models in terms of topic coherence.

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
