# A Implementation details

In this section, we include the hyperprameter details we use in this work, e.g. learning rate, batch size, etc. We apply different sets of hyperparameters, with respect to the dataset the neural topic model is trained on.

Table 1: Hyperparameter details

|  | **20NG** | **IMDb** | **Wiki** |
|---|---|---|---|
| **Learning rate** | 0.002 | 0.001 | 0.002 |
| **Batch size** | 200 | 200 | 500 |
| $k$ | $\{1, 5, 10, 15, 25, 30\}$ | $\{1, 5, 10, 15, 25, 30\}$ | $\{1, 5, 10, 15, 25, 30\}$ |

# B Contrastive loss derivation

We provide the proof of the inequality (**??**) in this section.

**Theorem 1.** *Let* $\mathbf{x}$ *denote the word count representation of a document,* $\mathbf{x}^+, \mathbf{x}^-$ *denote the positive sample and negative sample with respect to* $\mathbf{x}$*,* $f_\theta : \mathbb{R}^V \to \mathbb{R}^T$ *denote the mapping function of the encoder,* $\alpha \geq 0$ *denote the positive KKT multiplier, and* $\epsilon \geq 0$ *denote the strength of constraint. Suppose* $\beta = exp(\alpha)$*, then we have the following inequality*

$$\mathbb{E}_{\mathbf{x} \sim \mathcal{X}} \left[ \log(\exp(z \cdot z^+)) - \alpha \cdot (\log(\exp(z \cdot z^-)) - \epsilon) \right] \geq \mathbb{E}_{\mathbf{x} \sim \mathcal{X}} \left[ \log \frac{\exp(z \cdot z^+)}{\exp(z \cdot z^+) + \beta \cdot \exp(z \cdot z^-)} \right] \tag{1}$$

*Proof.* We rewrite the LHS in (1)

$$\mathbb{E}_{\mathbf{x} \sim \mathcal{X}} \left[ \log(\exp(\mathbf{z} \cdot \mathbf{z}^+)) - \alpha \cdot (\log(\exp(\mathbf{z} \cdot \mathbf{z}^-)) - \epsilon) \right] = \mathbb{E}_{\mathbf{x} \sim \mathcal{X}} \left[ \log(\exp(\mathbf{z} \cdot \mathbf{z}^+)) - \alpha \cdot (\log(\exp(\mathbf{z} \cdot \mathbf{z}^-))) + \alpha \cdot \epsilon \right]$$

$$\geq \mathbb{E}_{\mathbf{x} \sim \mathcal{X}} \left[ \log \left( \frac{\exp(\mathbf{z} \cdot \mathbf{z}^+)}{\beta \cdot \exp(\mathbf{z} \cdot \mathbf{z}^-)} \right) \right] \quad \text{as } \alpha, \epsilon \geq 0$$

$$\geq \mathbb{E}_{\mathbf{x} \sim \mathcal{X}} \left[ \log \left( \frac{\exp(\mathbf{z} \cdot \mathbf{z}^+)}{\exp(\mathbf{z} \cdot \mathbf{z}^+) + \beta \cdot \exp(\mathbf{z} \cdot \mathbf{z}^-)} \right) \right] \quad \text{as } \exp(\mathbf{z} \cdot \mathbf{z}^+) > 0$$

At this point, we conclude our proof. $\square$

# C Versions of loss function

We provide the description of versions of loss functions we use in this work.

**Contrastive approach - Using both positive and negative samples**

$$\mathcal{L}(-\mathbf{x}, \theta, \phi) = \mathbb{E}_{\mathbf{z} \sim q(\mathbf{z}|\mathbf{x})} \left[ -\log(p_\theta(\mathbf{x}|\mathbf{z})) + \mathbb{KL}(q_\theta(\mathbf{z}|\mathbf{x})||p(\mathbf{z})) \right]$$
$$- \mathbb{E}_{\mathbf{z} \sim q(\mathbf{z}|\mathbf{x})} \left[ \log \frac{\exp(\mathbf{z} \cdot \mathbf{z}^+)}{\exp(\mathbf{z} \cdot \mathbf{z}^+) + \beta \cdot \exp(\mathbf{z} \cdot \mathbf{z}^-)} \right] \tag{2}$$

**Contrastive approach - Using only positive sample**

$$\mathcal{L}(\mathbf{x}, \theta, \phi) = \mathbb{E}_{\mathbf{z} \sim q(\mathbf{z}|\mathbf{x})} \left[ -\log(p_\theta(\mathbf{x}|\mathbf{z})) + \mathbb{KL}(q_\theta(\mathbf{z}|\mathbf{x})||p(\mathbf{z})) \right] - \mathbb{E}_{\mathbf{z} \sim q(\mathbf{z}|\mathbf{x})} \left[ \mathbf{z} \cdot \mathbf{z}^+ \right] \tag{3}$$

**Contrastive approach - Using only negative sample**

$$\mathcal{L}(\mathbf{x}, \theta, \phi) = \mathbb{E}_{\mathbf{z} \sim q(\mathbf{z}|\mathbf{x})} \left[ -\log(p_\theta(\mathbf{x}|\mathbf{z})) + \mathbb{KL}(q_\theta(\mathbf{z}|\mathbf{x})||p(\mathbf{z})) \right] + \alpha \cdot \mathbb{E}_{\mathbf{z} \sim q(\mathbf{z}|\mathbf{x})} \left[ \mathbf{z} \cdot \mathbf{z}^- \right] \tag{4}$$

Figure 1: The influence of number of tokens chosen to construct random samples

## D Understanding number of chosen tokens

We demonstrate the effect of changing the number of tokens chosen for sampling. We perform training with different choices of $k \in \{1, 5, 10, 15, 20, 25, 30\}$ and record the topic coherence. For visibility, we normalize them to one common scale before plotting them in Fig 1. It can be seen that the performance initially increases as we select more tokens from the reconstructed output to substitute for the drawn sample. However, when the number of selected tokens $k$ grows too large, the topic coherence measure starts decreasing as $k$ increases. We hypothesize that the overwhelming number of substituted values can alter the semantic of the positive samples, while producing random negative sample.