# OpenReview forum: "Contrastive Learning for Neural Topic Model"
_NeurIPS.cc/2021/Conference — NeurIPS 2021 Poster_

### Official Review · Reviewer_u4uL · 2021-07-14

**Rating:** 7
**Confidence:** 3

**Summary:**

The paper proposes a novel approach to re-formulate discriminative goals as an optimization problem and designs a novel sampling method that facilitates the integration of external variables. It is a strong approach that outperforms other state-of-the-art neural topic models in three common benchmark datasets.

**Ethical Concerns:**

This paper has no ethical concern.

**Limitations And Societal Impact:**

The paper do not address the limitations and societal impact.

**Main Review:**

This paper explores a challenging problem and proposes a novel and intuitive approach for topic modeling. The paper is well-written and well-motivated. I think this paper may attract lots of researchers. Some minor issues should be addressed. For example, in Figure 3, the yellow and blue bars should have a legend.  In Figure 2, I recommend the authors add some descriptions of sampling in the caption.

**Time Spent Reviewing:**

1 hour

---

> ### Author Response · Authors · 2021-08-10
> **Response to Reviewer u4uL**
>
> Dear Reviewer,
>
> Thank you for your insightful comments and feedback! We are delighted that you find our paper interesting. Here are our answers to your detailed questions and comments.
>
> *1. Some minor issues should be addressed. For example, in Figure 3, the yellow and blue bars should have a legend. In Figure 2, I recommend the authors add some descriptions of sampling in the caption.*
>
> Regarding our minor issues, we thank the reviewer for pointing them out and we will definitely correct them in the revised version of the paper.

---

### Official Review · Reviewer_deqd · 2021-07-15

**Rating:** 7
**Confidence:** 4

**Summary:**

In this paper, the authors propose a novel contrastive objective for the neural topic model, which models the relations among samples without relying on the generative-discriminative architecture. In this work, they reveal that capturing the mutual information between the prototype and its positive samples provides a strong foundation for constructing coherent topics; while differentiating the prototype from the negative samples plays a less important role. Motivated by human behavior when comparing different documents, they also develop a novel sampling strategy to generate positive and negative samples under neural topic model setting, producing informative samples compared with other sampling strategies. Their experimental results show that their proposed method outperforms other state-of-the-art neural topic models in terms of topic coherence.

**Ethics Review Area:**

["I don’t know"]

**Limitations And Societal Impact:**

I wonder if other methods of measuring the distance between z in Eq 4 are available. It may be better to report the efficiency or the complexity of the proposed sampling strategy. Some references need to be clarified as they have been accepted by a conference or a journal but have still been linked with an Arxiv version, such as “Topic modeling with wasserstein autoencoders” .

**Main Review:**


This approach aims to learn document representations in the neural topic model as a contrastive objective and proposes a novel sampling strategy for contrastive learning, which is novel and interesting to me. The claims are well supported by experimental results. The paper is well written and easy to follow. And the results are moderately important.

**Time Spent Reviewing:**

5

---

> ### Author Response · Authors · 2021-08-10
> **Response to Reviewer deqd**
>
> Dear Reviewer,
>
> Thank you for your constructive comments and feedback! We are happy that you find our paper interesting. Here are our answers to your detailed questions and comments.
>
> *1. I wonder if other methods of measuring the distance between z in Eq 4 are available.*
>
> Regarding the distance between **z** in Eq 4, we have experimented with other measures such as Euclidean distance, Manhattan distance, and Cosine Similarity with number of topics T = 50 and reported the performance as follow:
>
> Distance | IMDb | 20NG | Wiki
>
> Manhattan | 0.195 ± 0.003 | 0.331 ± 0.002 | 0.496 ± 0.010
>
> Euclidean | 0.196 ± 0.004 | 0.330 ± 0.003 | 0.494 ± 0.011
>
> Cosine Similarity | 0.195 ± 0.005 | 0.334 ± 0.005 | 0.496 ± 0.014
>
> Dot product | 0.197 ± 0.006 | 0.334 ± 0.004 | 0.497 ± 0.009
>
> It can be seen that the results produced by using other distances are similar to dot product with negligible difference.
>
> *2.  It may be better to report the efficiency or the complexity of the proposed sampling strategy.*
>
> Regarding the efficiency or complexity of our proposed sampling method, it is worth noting that tf-idf representations along with top-k largest/smallest tokens of each vector are pre-computed simultaneously with bag-of-word input vectors. As a result, at the training time, we only need to extract top-k indices of the entries to substitute the reconstructed values, which nearly takes O(1) time complexity. Since this only requires little additional space cost for storing top-k indices, we are able to outperform state-of-the-art approaches with only additional marginal cost for our proposed sampling strategy.
>
> *3. Some references need to be clarified as they have been accepted by a conference or a journal but have still been linked with an Arxiv version, such as “Topic modeling with wasserstein autoencoders”.*
>
> Regarding the clarification of references, we thank the reviewer for the kind suggestion and we will definitely correct them in the revision.

---

### Official Review · Reviewer_YwvF · 2021-07-16

**Rating:** 5
**Confidence:** 3

**Summary:**

This paper proposed a Neural Topic Model based on contrastive learning, which considers both effects of positive and negative pairs. They proposed a sampling strategy that takes inspiration from human behavior in differentiating documents. Experimental results show that it outperforms other state-of-the-art neural topic models in terms of topic coherence.


**Ethical Concerns:**

This work has no ethical concerns.

**Ethics Review Area:**

["I don’t know"]

**Limitations And Societal Impact:**

The authors did not discuss the limitations, but this work has no potential negative societal impact.

**Main Review:**

Clarity:
This paper is clearly written and well organized.

Originality:
This work is a combination of well-known techniques, e.g., contrastive learning. They proposed a novel sampling strategy that takes inspiration from human behavior, to help neural topic models learn more meaningful representations. This work is simple and easy-to-follow.

Quality&Significance:
This work is technically sound, and all claims are well supported by both theoretical analysis and experimental results. However, the improvements over the baseline methods are small.

- My biggest concern is that the bag-of-word models may not consider the sequence information of the words in the document, so it poses a big challenge to the promotion of this model.
- From table 3, it is interesting that the positive sampling contributed more improvements. Does it mean that the original model is not good enough to identify redundant and uncritical information?

**Time Spent Reviewing:**

About 1 hour.

---

> ### Author Response · Authors · 2021-08-09
> **Response to Reviewer YwvF**
>
> Dear Reviewer,
>
> Thank you for your insightful comments and feedback! We are delighted that you find our paper interesting. Here are our answers to your detailed questions and comments.
>
> *1. My biggest concern is that the bag-of-word models may not consider the sequence information of the words in the document, so it poses a big challenge to the promotion of this model.*
>
> Regarding the limitation in sequence information of bag-of-word models, previous work of Hoyle et al., 2020 (arxiv.org/abs/2010.02377) has proven that bag-of-word neural topic model can also learn and encode contextualized information of Transformer-based models by applying knowledge distillation with BERT as the teacher model and neural topic model as the student. Our approach is orthogonal to their work, thus combining their approach with ours will be our future research direction.
>
> *2. From table 3, it is interesting that the positive sampling contributed more improvements. Does it mean that the original model is not good enough to identify redundant and uncritical information?*
>
> Regarding the contribution of positive sampling, it has been shown by previous works (Bachman et al., 2019 (arxiv.org/abs/1906.00910); Sun et al., 2020 (arxiv.org/abs/1908.01000); Park et al., 2020 (arxiv.org/abs/2007.15651)) that comparison among positive samples helps the model recognize mutual information that enhances the ability to capture high-level features from the input. In the case of neural topic models, the features are the topics among the documents. As a result, for the phenomenon that positive sampling contributed more improvements, we hypothesize that the original model is inefficient in modeling mutual high-level features among documents, which has been proven in (Zhu et al., 2020) that the amount of such information is low in adversarial training frameworks of Adversarial Topic Model.

---

### Official Review · Reviewer_5i58 · 2021-07-17

**Rating:** 6
**Confidence:** 3

**Summary:**

The paper proposes a contrastive learning framework to improve the performance of neural topic model in terms of topic coherence. To guide the model focus on the negative samples and separate topics more clearly, the paper proposes an adaptive scheduling strategy to estimate $\beta$. And a word-based sampling strategy is proposed to sample negative and positive samples, which is mainly based on the tf-idf scores.

**Limitations And Societal Impact:**

The authors have adequately addressed the limitations and potential negative societal impact of their work.

**Main Review:**

Strengths:
1. Using contrastive learning for neural topic model seems novel, and can address the main drawbacks of Adversarial Topic Model.
2. The experiments and analysis are comprehensive.

Weakness:
1. The improvement over traditional contrastive learning framework seems limited. And why the authors choose the triangle scheduling method? Does this schedular have better performance than others? Is there any theoretical or empirical analysis about why this scheduling method performs well?
2. The novelty of the sampling strategy seems limited. Are there any other statistical indicators for sampling besides tf-idf score? What is the novelty or improvements compared with other word-based sampling methods?
3. The performance gap through all epochs in Figure 4 seems similar all the time. Can the authors provide some analysis toward such phenomenon?


**Time Spent Reviewing:**

7 hours

---

> ### Author Response · Authors · 2021-08-09
> **Response to Reviewer 5i58**
>
> Dear Reviewer,
>
> Thank you for your constructive comments and feedback! We are happy that you find our paper interesting. Here are our answers to your detailed comments and questions.
>
> *1. The improvement over traditional contrastive learning framework seems limited.*
>
> Regarding the improvement against the traditional contrastive learning, we have conducted statistical tests to evaluate the statistical significance of our improvement on three datasets 20NG, IMDb, and Wiki, then reported the p-values as follows
>
> Metric | 20NG - T = 50 | 20NG - T = 200 | IMDb - T = 50 | IMDb - T = 200 | Wiki - T = 50 | Wiki - T = 200
>
> p-value | 0.0145 | 0.0494 | 0.0291 | 0.0012 | 0.0344 | 0.0156
>
> As it can be seen, all of the p-values are smaller than 0.05, which proves the statistical significance in the improvement of our method against traditional contrastive learning.
>
> *2. Why the authors choose the triangle scheduling method? Does this schedular have better performance than others? Is there any theoretical or empirical analysis about why this scheduling method performs well?*
>
> Regarding the scheduling method, we estimated the derivative of the objective function with respect to **z** in Eq. 5 in order to approximate the gradient and its correlation to $\beta$. As it can be seen that the magnitude of the gradient is proportional to $\beta$, we decide to employ triangular scheduling for $\beta$ since the parameter update should be controlled carefully at the early and the late stage of the training procedure (Luo et al., 2020 (arxiv.org/abs/2010.06351)). However, for further empirical evidence, we also conducted the experiments with other scheduling methods, i.e., decaying (You et al., 2019 (arxiv.org/abs/1908.01878)), and constant schedule with warm-up (Zagoruyko et al., 2016 (arxiv.org/abs/1605.07146)), with the value of k = 15 and T = 50 on IMDb, 20NG, and Wiki dataset, and reported the results (NPMI) as follows:
>
> Method | IMDb | 20NG | Wiki
>
> Triangular | 0.197 ± 0.006 | 0.334 ± 0.004 | 0.497 ± 0.009
>
> Decaying | 0.185 ± 0.004 | 0.318 ± 0.005 | 0.467 ± 0.011
>
> Constant with warm-up | 0.183 ± 0.003 | 0.311 ± 0.001 | 0.462 ± 0.008
>
> As it can be seen, the triangular schedule method consistently outperforms other scheduling approaches.
>
> *3. The novelty of the sampling strategy seems limited. Are there any other statistical indicators for sampling besides tf-idf score? What is the novelty or improvements compared with other word-based sampling methods?*
>
> Regarding the proposed sampling strategy, although TF-IDF is a simple sampling method, it has been proven effective in solving simple problems such as choosing important/non-important words in a document, which is suitable for our task of filtering salient and unimportant tokens to draw positive and negative samples. To the best of our knowledge, we are the first to propose word-based sampling strategy for contrastive learning in neural topic models. To have a fair judgement, we also conduct experiments with two other complex sampling methods using Principal Component Analysis (PCA) or Singular Value Decomposition (SVD). Specifically, we decompose the reconstructed and original input vectors into singular values and then replace the largest/smallest singular values of the input with the largest/smallest ones of the reconstructed to obtain negative/positive samples, respectively. For SVD, we choose k = 15 largest/smallest values for substitution whereas for PCA, we decompose the input vector onto 50-d space in order to make it similar to the latent space of neural topic model (number of topics T = 50) and proceed to substitute k = 15 largest/smallest values as in SVD. We conducted our experiments on 3 datasets IMDb, 20NG, and Wiki with T = 50, and reported the results (NPMI) as follows:
>
> Sampling method | IMDb | 20NG | Wiki
>
> PCA | 0.184 ± 0.004 | 0.325 ± 0.003 | 0.481 ± 0.005
>
> SVD | 0.181 ± 0.004 | 0.313 ± 0.003 | 0.476 ± 0.014
>
> TF | 0.196 ± 0.003 | 0.332 ± 0.006 | 0.495 ± 0.008
>
> IDF | 0.193 ± 0.001 | 0.334 ± 0.004 | 0.490 ± 0.009
>
> TF-IDF | 0.197 ± 0.006 | 0.334 ± 0.004 | 0.497 ± 0.009
>
> As it can be obviously seen, despite its simplicity, TF-IDF-based sampling method outperforms other complicated sampling methods in our tasks.
>
> *4. The performance gap through all epochs in Figure 4 seems similar all the time. Can the authors provide some analysis toward such phenomenon?*
>
> Regarding the similar performance gap denoted in Figure 4, we hypothesize that the gap is caused by the limited ability of the constant schedule to adapt for the training dynamics of neural topic model (Xu et al., 2019 (arxiv.org/abs/1909.09712)), likely due to the insufficiency or redundancy of the gradient used to update the parameters during back-propagation. Interestingly, our finding in the performance gap is also consistent with previous work that utilizes adaptive schedule (Luo et al., 2020 (arxiv.org/abs/2010.06351))

---

### Author Response · Authors · 2021-08-10
**Overall Response to All the Reviewers**

We would like to thank all reviewers for the positive assessment and thorough comments. We have revised our paper based upon the feedback and provided individual responses for each reviewer.

---

### Decision · Program_Chairs · 2021-09-27

**Decision:**

Accept (Poster)

**Comment:**

This paper tackles neural topic model training and proposes a new sampling strategy, altogether improving performance under NPMI, topic coherence and downstream classification performance.

The reviewers find the paper interesting, novel, and intuitive, with some concerns about design choices. The authors answered the reviewers questions thoroughly with some additional ablations that help justify the design choices, e.g., the tf-idf sample weighting strategy. The authors should include these additional results in the paper.
I would also recommend some polishing of the presentation: for example in Fig 3 the fonts are too small and the bars on the right side are not labelled; could that figure not be a table? Additionally, tf-idf should not be rendered in math mode (surrounded by \$); instead, use \\emph.